

# Seed bank and growth comparisons of native (*Virgilia divaricata*) and invasive alien (*Acacia mearnsii* and *A. melanoxylon*) plants: implications for conservation

Stefan A. Goets[1,2], Tineke Kraaij[1] and Keith M. Little[1]

[1] School of Natural Resource Management, Nelson Mandela University, George, South Africa
[2] South African Environmental Observation Network, Kimberley, South Africa

Corresponding author
Tineke Kraaij,
tineke.kraaij@mandela.ac.za

## ABSTRACT

**Background**. Invasive alien plants with long-lived dormant seed banks and fast growth rates are difficult to manage. *Acacia mearnsii* and *Acacia melanoxylon* are two such invaders in the southern Cape of South Africa which occasionally co-occur with a native, ecologically analogous species, *Virgilia divaricata*. We compared the performance of these three species to determine potential for the native species to be used in management of the invasives.

**Methods**. We compared the study species in terms of (i) soil seed bank densities, their vertical distribution, and the viability of seeds underneath the canopies of mature trees; (ii) seedling growth from planted seeds over a period of three months; and (iii) growth rates of saplings over a period of 10 months in stands that have naturally regenerated in the field (these stands were dominated by *A. mearnsii*) and where saplings have been exposed to varying levels of competition from surrounding saplings.

**Results**. Seed bank densities differed significantly among species but not among soil depth classes. *Acacia mearnsii* had the highest seed bank densities (mean of 7,596 seeds m$^{-2}$), followed by *V. divaricata* (938 seeds m$^{-2}$) and *A. melanoxylon* (274 seeds m$^{-2}$). Seed viability was high (87–91%) in all three study species and did not differ significantly among species or soil depth classes. As seedlings, *V. divaricata* significantly outgrew *A. mearnsii* in terms of height, root and shoot dry mass, and root:shoot ratio. Relative growth (the relationship between growth in height and initial height) was negative in the seedlings of both species. Trends during the sapling stage were opposite to those during the seedling stage; *A. mearnsii* (but not *A. melanoxylon*) saplings significantly outgrew *V. divaricata* saplings in height, while relative growth rates were positive in all species. Sapling growth of all species was furthermore uninfluenced by the collective biomass of surrounding competitors.

**Discussion**. Our findings suggest that amongst the measures considered, *A. mearnsii*'s success as an invader is primarily attributable to its large seed banks, and secondly to its vigorous growth in height as saplings. However, the superior growth performance of *V. divaricata* seedlings and no apparent negative effect of competition from the acacias on sapling growth show promise for its use in integrated management of the acacias.

## INTRODUCTION

Competitive performance is an important factor for determining the success or failure of plant invasions and accordingly native-invasive plant competition has been studied extensively (*Daehler, 2003*; *Gioria & Osborne, 2014*; *Levine, Adler & Yelenik, 2004*; *Theoharides & Dukes, 2007*; *Vilà & Weiner, 2004*). Review studies suggested that the performance of native plants may be equal or superior to that of invasive alien plants (IAPs) under certain conditions (*Daehler, 2003*; *Funk et al., 2008*), but emphasized the importance of using comparable plant life forms in pair-wise competition experiments (*Gioria & Osborne, 2014*). Native species that show superior performance could potentially aid in the management of invasive species (*Vilà & Weiner, 2004*). Accordingly, habitat management (also termed 'cultural control'), often used in agriculture or forestry (*Leihner, 2002*), involves the alteration of a habitat to favour the target species over that of unwanted species (such as IAPs) and includes the use of native species to increase the amount of competition experienced by the unwanted species (*Wittenberg & Cock, 2001*). Using habitat management as part of an integrated pest management strategy may require high initial investment, but tends to require fewer follow up treatments and rehabilitation may be particularly successful when used in conjunction with other control methods such as biological agents (*Hatcher & Melander, 2003*).

Seed ecology, and specifically seed bank density and longevity, are important determinants of plant competitive performance (*Richardson & Kluge, 2008*). Seed dormancy allows a species to build up propagule pressure over an extended period of time, increasing its likelihood of survival (*Porsild, Harrington & Mulligan, 1967*). Invasive plant species with dormant seed banks are particularly difficult to manage since, even after the adult plants are removed, they can recruit *en masse* if germination requirements are met (*Goets, Kraaij & Little, 2017*; *Marchante, Freitas & Hoffmann, 2010*; *Richardson & Kluge, 2008*). Divergent germination requirements and timing may also influence competitive interactions among species (*Finch-Savage & Leubner-Metzger, 2006*; *Gioria & Osborne, 2014*; *Parker, 1989*). Inherent growth rate is another important determinant of plant success, whereby rapid initial growth is associated with superior resource acquisition rates, thus enabling exclusion of other, less competitive, species (*Craine, 2009*). The effect of competition may be a reduction in the fitness (i.e., biomass production, flower production, seed production) of one or both of the species (*Te Beest et al., 2013*; *Vaugh & Young, 2015*). Fast growth rates are typically observed in pioneer species, and often account for invasive species being problematic. However, the performance of a native species with similar ecological traits may compare favourably with that of an invasive species in the same environment (*Daehler, 2003*; *McDowell & Moll, 1981*).

Intact native vegetation is known to present barriers to invasion (*Duncan, 2011*). By the same token, disturbances which reduce or remove competition from native vegetation

facilitate invasion by alien plants (*Hobbs & Huenneke, 1992*). This suggests that native plant species may potentially be useful in the management of invasive species. Although several studies have compared competitive performance (in terms of seed banks, growth rates and other aspects) between native and alien species pairs (see reviews by *Daehler, 2003*; *Funk et al., 2008*; *Vilà & Weiner, 2004*), few of these have applied their findings to develop invasive plant management strategies that make use of native plant species. However, restoration studies commonly consider the value of native plants, including their soil-stored seed banks, to suppress invasives after clearing operations (*Bakker & Wilson, 2004*; *Taylor & McDaniel, 2004*; *Tererai et al., 2015*). Biological control, although it seldom involves plant-on-plant competition, also relies on exploiting negative interactions among biological organisms to suppress IAPs. Biocontrol agents often target seed production of invasive plants, such as seed boring weevils and flower-galling midges (*Adair, 2004*; *Impson et al., 2008*; *Impson, Hoffmann & Kleinjan, 2009*). These slow down the accumulation rate of seeds into seed banks, but the remaining seeds must be accounted for in subsequent management activities to prevent re-establishment of an invasive (*Tererai et al., 2015*; *Van Wilgen et al., 2016*).

*Acacia mearnsii* De Wild and *Acacia melanoxylon* R.Br. are two aggressive invaders in the southern Cape of South Africa and occasionally co-occur with a native, ecologically analogous species, *Virgilia divaricata* Adamson. All three species are fast-growing, nitrogen-fixing trees in the family Fabaceae. *Virgilia divaricata* and *A. mearnsii* are small to medium trees living for 20 to 40 years (*Coates Palgrave, Drummond & Moll, 2002*; *Coetsee & Wigley, 2013*; *Phillips, 1926*; *Searle, 1997*) and *A. melanoxylon* is a medium to tall tree living in excess of 100 years (*Louppe et al., 2008*). The two acacias originate from south-east Australia, *A. mearnsii* being a pioneer in woodland and forest and *A. melanoxylon* an overstory forest species (*Praciak et al., 2013*; *Searle, 1997*). Under natural conditions, *V. divaricata* typically occurs in dense monospecific stands along forest edges (forest-fynbos ecotone) after fire, acting as a pioneer for indigenous forest re-establishment (*Coates Palgrave, Drummond & Moll, 2002*; *Coetsee & Wigley, 2013*). The acacias are particularly invasive in disturbance-prone habitats such as riparian zones and forest edges (*Baard & Kraaij, 2014*; *Greet, 2016*; *Tererai et al., 2015*), a niche often occupied by *V. divaricata*. All three species have soil-stored seeds which may remain dormant for up to 50 years (*Geldenhuys, 1994*; *Harrington, 1972*; *Seydack, 2002*). The seeds require scarification in order to germinate, which is provided by the heat of fire (*Jeffery, Holmes & Rebelo, 1988*; *Searle, 1997*; *Louppe et al., 2008*; *Goets, Kraaij & Little, 2017*), or ingestion by birds in the case of *A. melanoxylon* (*Richardson & Kluge, 2008*). Similarities in germination requirements and timing among the three species suggested that germination requirements and timing are unlikely to confer meaningful competitive advantages to one species over the other (*Goets, Kraaij & Little, 2017*). *Virgilia divaricata* has the largest seed size of the three species (∼6 mm long; *Phillips, 1926*), with seeds of both acacias being 3–5 mm long (*Louppe et al., 2008*; *Praciak et al., 2013*). The seeds of *V. divaricata* are not known to be dispersed by biotic agents, whereas those of *A. mearnsii* are dispersed over short distances by ants and *A. melanoxylon* by birds and water (*Greet, 2016*; *Praciak et al., 2013*). Seeds of all three species do not require light in order to germinate (*Goets, Kraaij & Little, 2017*), however, their seedlings are shade-intolerant (*Louppe et al., 2008*; *Phillips, 1926*; *Searle, 1997*). The three species can

thus co-occur within the distribution range of *V. divaricata* wherever disturbance has taken place and where seeds are present. Disturbances commonly associated with these species are fires and plantation forestry operations (*Baard & Kraaij, 2014*; *Kraaij, Cowling & Van Wilgen, 2011*).

In this study, we compared aspects of seed ecology and growth performance between two IAP species (*A. mearnsii* and *A. melanoxylon*) and a native ecological analogue (*V. divaricata*). In particular, we asked whether the native species may outperform the invasive species in terms of their (i) seed bank densities and viability, and (ii) growth rates both as seedlings and competitively as saplings. We aimed to provide a preliminary screening of the potential for using the indigenous species in integrated management of the invasive species.

## MATERIALS AND METHODS

### Study sites

The study area was the southern Cape coast of South Africa between the towns of George (−33.97°, 22.75°) and Nature's Valley (−33.97°, 23.55°). The climate of the study area is temperate with an average of 800–1,078 mm of year-round rainfall (peaking in April and October) and daily summer and winter temperatures averaging 20 °C and 12 °C, respectively (*Tyson & Preston-Whyte, 2000*). The underlying geology mostly comprises rocks derived from sandstone, quartzite, and shale forms of the Table Mountain Group (*Schafer, 1992*). All the study sites were situated on the coastal plateau, with mostly colluvial and alluvial soils with yellow-brown to grey-brown colours and sandy loam to silty loam textures (*Schafer, 1992*). The major soil groups of the area are duplex (Estcourt and Klapmuts), hydromorphic (Longlands and Katspruit), and podzols (Lamotte and Witfontein) (*Soil Classification Working Group, 1991*). The vegetation of the area comprises fire-sensitive Southern Afrotemperate forest (*Mucina & Geldenhuys, 2006*) and fire-prone fynbos shrubland (*Rebelo et al., 2006*). Due to the history of disturbance associated with extensive commercial plantations in the area (*Kraaij, Cowling & Van Wilgen, 2011*), high densities of *A. mearnsii* and to a lesser extent *A. melanoxylon* occur among indigenous fynbos and forest vegetation (*Baard & Kraaij, 2014*). These invaders are known to alter natural disturbance regimes such as fires, floods, and forest gap formation, which may further promote their proliferation (*Kraaij et al., 2013a*; *Midgley et al., 1990*; *Russell & Kraaij, 2008*).

### Seed banks

Seed banks were sampled from underneath canopies of 30 mature trees (10 trees per species) dispersed within the study area (Supplemental Information 1). Approval was obtained from South African National Parks and MTO Forestry to conduct field studies on land under their jurisdiction. Trees were selected based on criteria regarding stem diameter (20–50 cm at 50 cm above ground level), slope (<30°), canopy overlap with conspecifics (<30%), and habitat type. The study species characteristically occur in four habitat types, namely Southern Afrotemperate forest, fynbos shrubland, forest-fynbos ecotone, and severely disturbed or transformed land (*Baard & Kraaij, 2014*; *Coetsee & Wigley, 2013*).

We aimed for comparable representation of habitat types among the study species but did not achieve a perfectly balanced sampling design in terms of this factor (Supplemental Information 1). For each sampled tree, soil core samples were taken at predetermined sampling locations (at 45 and 90 cm from the stem) along four radiating transect lines. These four lines were positioned at 90° intervals, starting at 45° with 0° directed upslope. This was to account for potential seed drift down slopes, although seed density did not differ significantly up-slope *vs.* down-slope of stems (paired *t*-test $t_{(29)} = 1.79$, $p = 0.084$). A core sampler with a diameter of 4.5 cm and 20 cm depth was used to take samples. At each sampling location, six litter samples were collected from an area equivalent to the cross-section of the corer, followed by six cores from the surface to 5 cm ('shallow') and two cores from 5–15 cm ('deep'). Our depth classes followed those of *Auld & Denham (2006)* where 0–5 cm represented 'effective germination depths' and the zone where fire may destroy seeds (*Pieterse & Boucher, 1997*; *Pieterse & Cairns, 1987*), whereas germination is possible but rare from depths of 15 cm (*Richardson & Kluge, 2008*). Seeds were extracted from litter and soil samples by sieving out fine material and sorting the remaining material by hand, and thereafter counted. Seed bank density data comprised of seed counts for each soil depth class, and combined for the eight sample plots per tree, resulting in ten replicates per species × depth class combination. The measure of interest was number of seeds per unit of ground surface area for each of the depth layers as that translates to potential density of recruits which is the aspect of interest in IAP management. This measure also enabled comparison of observed seed bank densities with those reported in the literature.

A tetrazolium-chloride stain test (*Porter, Durrell & Romm, 1947*) was used to determine the proportional viability of seeds from the different species and depth classes. Seeds were cut longitudinally, soaked overnight in a 2% 2,3,5-triphenyl-tetrazolium chloride solution, and visibly stained seeds enumerated. Ten to 50 seeds (depending on availability from samples) were tested per habitat type per species × depth class combination, aiming for a total of 100 seeds per species × depth class combination. In some habitat types, no seeds were obtained in particular species × depth classes, which reduced the number of samples for statistical analysis. Seed viability was expressed as a proportion of viable seeds from the total number of seeds tested per replicate of species × depth class combination.

## Seedling growth

Seeds of *V. divaricata* and *A. mearnsii* (1,500 seeds species$^{-1}$) were subjected to a smoke and heat treatment to facilitate germination (*Goets, Kraaij & Little, 2017*), after which the seeds were planted at the beginning of the cold season (June 2014) within an enclosed (to exclude large herbivores) plot on the George Campus of Nelson Mandela University (−33.963833°, 22.533333°). Note that the seedling growth study was part of a pilot study, and *A. melanoxylon* had not been considered as a study species at that time. The seeds were planted 1 cm deep, with the two species arranged randomly and spaced 5 cm apart for seedlings so as to be unaffected by competition (*Midgley, Hoekstra & Bartholomew, 1989*). The plot received full sun until late afternoon, as would occur following fire. Above-ground heights of germinants ('seedlings') were measured one ($T_1$) and three ($T_2$) months after planting. Seedlings were harvested after the second measurement, and dried to a constant

mass in paper bags at 60 °C for 48 h. Above and below-ground parts (hereafter referred to as 'shoots' and 'roots', respectively) were weighed separately to determine their respective dry masses.

## Sapling growth

Circular study plots (2 m radius) were located where at least one individual of *V. divaricata* (considered as reference species) occurred interspersed within stands of co-occurring *A. mearnsii* and *A. melanoxylon* and little other vegetation (Fig. 1). Eighteen plots were situated at locations ($-33.883133°$, $22.877583°$; and $-33.88635°$, $23.011617°$) within the Garden Route National Park where pine plantations were clear-felled three to five years prior to the study, and where saplings subsequently attained sizes of 0.2–8.0 cm groundline diameter (GLD; stem diameter at 8 cm above ground level). Saplings of such sizes are ideal for growth studies in displaying rapid (measurable) growth over a short period (*Kozlowski, 1971*). *Acacia mearnsii* considerably outnumbered the other two species in all the chosen study plots as we were unable to find sites containing more equitable numbers of saplings of the three species. Saplings of all three study species, with heights greater than 50 cm, were marked and numbered. The first measurements ($T_1$) were taken in August–September 2015, and the second ($T_2$) in June 2016. Measurements included GLD (cm), height (cm), and distance from plot centroid (cm), with a total of 1,252 saplings (34 of *V. divaricata*; 1,138 of *A. mearnsii*; 80 of *A. melanoxylon*) surveyed. For all marked saplings in the study plots, absolute growth increment ($T_2–T_1$) was calculated for height, GLD, and biomass index 'BI' (calculated as GLD squared multiplied by height; *Eccles, Kritzinger & Little, 1997*).

To assess competitive performance of individual saplings (of the three study species) in relation to the competition collectively exerted by nearby saplings, a subset of marked saplings was first identified as focal saplings. These focal saplings were all the saplings (21 for *V. divaricata*; 331 for *A. mearnsii*; 33 for *A. melanoxylon*) that occurred within the central 1 meter radius plot of the (2 meter radius) study plot (Fig. 1). The performance of each focal sapling was assessed in relation to the collective competition exerted by the remainder of saplings within the two meter radius plot. Plot competition was expressed as the total BI of all saplings (excluding the focal sapling), by summing the average BI of each sapling between $T_1$ and $T_2$ (a measure of saplings' average biomass during the study period). Relative growth rate in BI $\left(\frac{T_2-T_1}{T_1}\right)$ was used as the measure of growth in focal saplings, since the relationships between BI increment and initial BI did not differ among species (investigated previously).

## Statistical analyses
### Seed bank

To compare seed bank densities among soil depth classes, we multiplied the number of seeds extracted from the deep layer (where two cores were sampled per tree as opposed to six in each of the shallow and litter layers) by a factor of three. Conversion to seeds $m^{-2}$ was done per sample area by multiplying the seed count with a factor of 104.79 (10,000 $cm^2$ divided by 95.42 $cm^2$, which is the surface area of six cores or one sample area per tree), and then averaged per tree. Due to non-normality the data were $\log(x+1)$ transformed (*Barlett,*
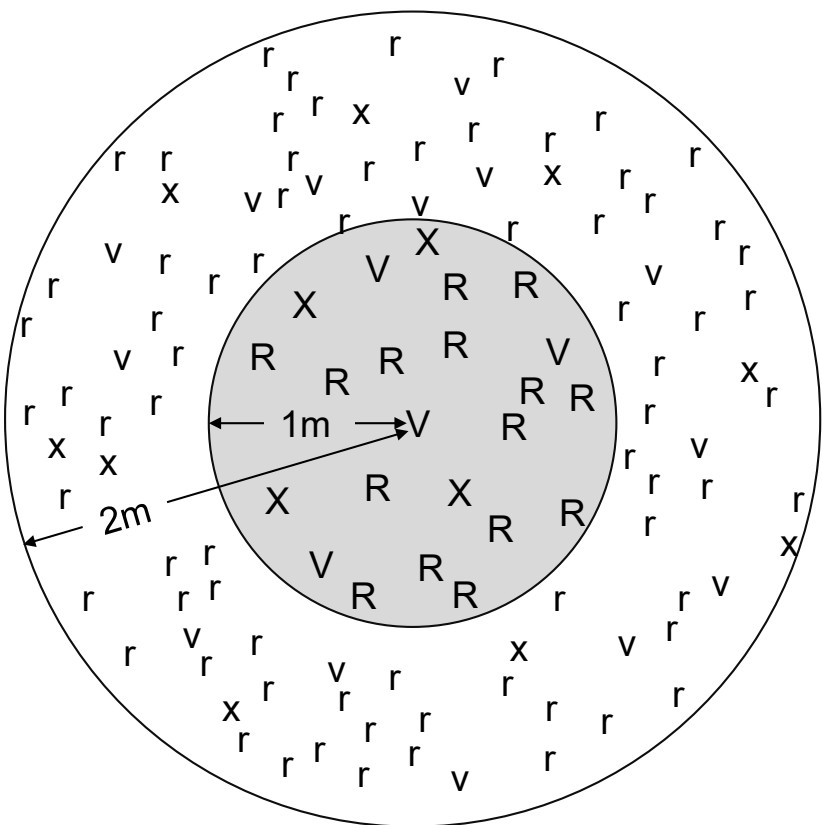

**Figure 1** **Schematic of plot layout pertaining to sapling growth assessments.** Hypothetical sampling plot where growth of all saplings (i.e., occurring in the inner and outer circles) with height > 50 cm of *Virgilia divaricata* ('V' and 'v'), *Acacia mearnsii* ('R' and 'r') and *A. melanoxylon* ('X' and 'x') were measured. Additionally we assessed the effect of competition exerted by nearby saplings on the growth of focal saplings individually. For this assessment, each focal sapling (V, R, or X) occurred within the inner one meter radius circular plot (shaded), and each focal sapling's growth increment was related to the summed biomass (expressed as an index, see text) of all the remaining saplings (V, R, X, v, r, and x) within the remainder of the 1 meter radius (shaded) and 2 meter radius (unshaded) circular plots combined.

*1947*). We used mixed-model analysis of variance (ANOVA) (*Littell, 2002*) to assess the effects on seed bank density of species and depth class (as fixed effects), habitat type (as random effect), and second-order interactions among these (as fixed effects). The effects of species and depth class on seed viability were tested (representation across habitat types was too poor to enable inclusion of this factor). The viability expressed as proportions did not conform to assumptions of normality and homogeneity of variance and were subjected to a rank transformation. The ranked and unranked data were tested using a factorial ANOVA, yielding comparable results. Test results from the unranked data were regarded as valid, as the assumptions for ANOVA were reasonably satisfied (*Montgomery, 2013*).

### Seedling growth

Root and shoot dry masses, and calculated root:shoot ratios did not conform to assumptions of normality and homogeneity of variance, thus between species comparisons were

made using Mann–Whitney $U$ tests (using the $Z$-score; *Privitera, 2017*). Absolute height increment was calculated ($T_2$–$T_1$) and its relationship with initial height (at $T_1$) explored—this relationship constitutes 'relative growth rate'. To determine whether relative growth in *A. mearnsii* differed from that in *V. divaricata* we used a general linear model (GLM) with height increment as the dependent variable and initial seedling height and species (incorporated as dummy variable with *V. divaricata* as reference species) as explanatory variables.

### Sapling growth

GLMs were used to explore whether relative growth rates in terms of height, GLD, and BI differed between the acacias and *V. divaricata*, with initial size (at $T_1$) and species (as dummy variables with *V. divaricata* as reference species) as explanatory variables. To explore how relative growth rate in BI of focal saplings was affected by the extent of competition (∼plot biomass) and whether this differed among species, a GLM was conducted on the relative growth rate in BI of focal saplings as the dependent variable, and plot biomass and species (as dummy variables with *V. divaricata* as reference species) as explanatory variables. As the primary inquiry in this part of the study was in terms of differences in competitive growth among species, GLM slope coefficients (as opposed to intercept coefficients) relevant to species comparisons were the focus in interpretations (comprehensive GLM outputs were provided in Supplemental Information 2). All statistical analyses were carried out using STATISTICA v.13 (*TIBCO Software Inc., 2017*).

## RESULTS

### Seed banks

Seed bank densities differed significantly among species and among habitat types but not among soil depth classes (Table 1; Figs. 2A & 2B). *Acacia mearnsii* had the highest seed bank densities, followed by *V. divaricata* and *A. melanoxylon*. The average seed bank density in the sampled soil profile, for all three species, was 2,936 seeds m$^{-2}$, with large variation (from ∼100 to several thousand seeds m$^{-2}$) evident in all three species (Table 2). There was a significant interaction between species and habitat (Table 1) with seed densities being disproportionately high in *A. mearnsii* in disturbed habitat, and in *V. divaricata* in forest-fynbos ecotone and disturbed habitat (Fig. 2A). In terms of reliability of seed banks, *V. divaricata* and *A. mearnsii* appeared to maintain seed banks more consistently (with 22% and 20% of sample plots per depth class yielding no seed, respectively) than *A. melanoxylon* (63%). Seed viability was high in all three species (*A. mearnsii* 91%, *V. divaricata* 88%, and *A. melanoxylon* 87%) and did not differ significantly among species ($F_{(2,20)} = 0.83$; $p = 0.452$) or depth classes ($F_{(2,20)} = 3.01$; $p = 0.072$) (Fig. 2C).

### Seedling growth

Root dry mass ($Z = -11.767$; $p < 0.001$), shoot dry mass ($Z = -10.914$; $p < 0.001$), and root:shoot ratio ($Z = -9.625$; $p < 0.001$) of harvested seedlings were significantly higher in *V. divaricata* (average root mass 132 mg; shoot mass 98 mg; root:shoot ratio 1.48) than in *A. mearnsii* (8 mg; 15 mg; 0.63) (Fig. 3). *Virgilia divaricata* displayed a significant

**Table 1** **Mixed-model results of the effects of species, soil depth and habitat type on seed bank densities of *Virgilia divaricata*, *Acacia mearnsii* and *A. melanoxylon*.** Results of mixed-model analysis of variance to assess the effects of species and soil depth (as fixed effects), habitat type (as random effect), and second-order interactions (as fixed effects) on seed bank density, expressed as $\log(\text{seeds m}^{-2} + 1)$. See text for details of categories within factors.

| Effect | df Effect | MS Effect | df Error | F | p |
|---|---|---|---|---|---|
| Species | 2 | 23.08 | 6.29 | 17.65 | 0.002 |
| Depth | 2 | 1.32 | 66.00 | 2.88 | 0.063 |
| Habitat | 3 | 8.08 | 66.00 | 17.56 | <0.001 |
| Species*Depth | 4 | 0.35 | 66.00 | 0.76 | 0.551 |
| Species*Habitat | 6 | 1.92 | 66.00 | 4.18 | 0.001 |
| Depth*Habitat | 6 | 0.36 | 66.00 | 0.78 | 0.584 |

negative relationship between growth in height and initial height, with the relationship for *A. mearnsii* not differing significantly from *V. divaricata* (Figs. 4A & 4B; Supplemental Information 2). Growth rates of seedlings declined during the two to three months following germination. The average height of seedlings at three months was 5.3 cm and 1.0 cm for *V. divaricata* and *A. mearnsii* respectively, with maximum heights of 9.8 cm and 1.5 cm respectively.

### Sapling growth

Saplings of all three species displayed positive relationships between their growth increment and initial size (in terms of height, GLD, and BI, although not significant in terms of height in *V. divaricata* and GLD in *A. melanoxylon*; Fig. 4). The relationship between initial height and height increment was significantly more positive for *A. mearnsii* than for *V. divaricata* but it did not differ between *V. divaricata* and *A. melanoxylon* (Figs. 4C–4E; Supplemental Information 2). The positive relationship between initial GLD and GLD increment was significant for *V. divaricata* and did not differ from that of *A. mearnsii*, while being significantly more positive than that of *A. melanoxylon* (Figs. 4F–4H). The positive relationship between initial BI and BI increment was significant for *V. divaricata* and did not differ from the two acacias (Figs. 4I–4K). The extent of competition (summed competitor BI) had no effect on the growth (relative increase in BI) of focal saplings of any of the study species (Supplemental Information 2).

## DISCUSSION

### Seed banks

One of the most apparent differences among the study species related to seed bank densities with the average of *A. mearnsii* being almost one order of magnitude larger than *V. divaricata*, while seed densities in *A. melanoxylon* were the smallest. Our estimates of seed bank densities varied widely within species, with ranges observed in *V. divaricata* and *A. mearnsii* being largely consistent with previous investigations (Table 2), but substantially lower in *A. melanoxylon* than recorded by *Donald (1959)*. Generally seed banks are highly variable in space and time and are affected by various factors, *inter alia* climate, habitat type, geomorphology, population demographics, plant density, dispersal strategy, and seed

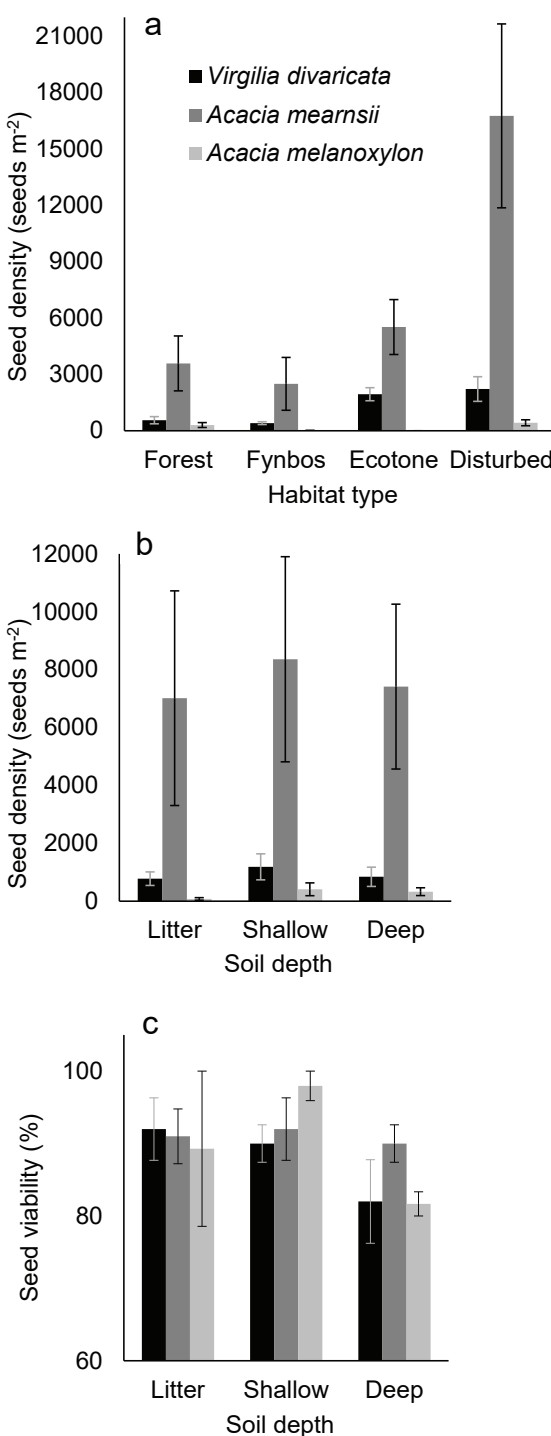

**Figure 2** **Seed bank densities in different (A) habitat types and (B) soil depth layers, and (C) percentage viability of these seeds of *Virgilia divaricata*, *Acacia mearnsii*, and *A. melanoxylon*.** Means ± SE of untransformed data are shown for seed bank densities and percentage viability. Seeds were sampled under the canopy of mature trees (with stem diameter of 20–50 cm) of three species (*N* = 10 trees per species) in natural stands along the southern Cape coast of South Africa.

**Table 2  Seed bank densities (seeds m$^{-2}$) and viability (%) of *Virgilia divaricata*, *Acacia mearnsii*, and *A. melanoxylon* recorded in this study and compared to seed bank densities reported in literature.**

| Species | Seed bank density (seeds m$^{-2}$)[h] | | Seed viability (%) | |
|---|---|---|---|---|
| | Current study mean (range) | Literature | Current study | Literature |
| *V. divaricata*[g] | 938 (170–3,406) | 15–685[e]; 500–2,000[f] | 88 | 90–99[f] |
| *A. mearnsii* | 7,596 (57–35,060) | 5,314[a]; 38,340[c] | 91 | 92[b] |
| *A. melanoxylon* | 274 (0–1,262) | 2,860–94,420[d] | 87 | 70[c]; 90[d] |

Notes.
[a] *Pieterse (1997)*.
[b] *Garner (2007)*.
[c] *Milton & Hall (1981)*.
[d] *Donald (1959)* cited in *Milton & Hall (1981)*.
[e] *Geldenhuys (1994)*.
[f] *Phillips (1926)*.
[g] *V. capensis* was also included in literature searches as *V. divaricata* and *Virgilia oroboides* were both referred to as *V. capensis* prior to taxonomic revision.
[h] To enable comparison with the literature, our seed counts were converted to seed numbers m$^{-2}$.

predation (*Parker, 1989*). Seed densities of *A. mearnsii* that have been disproportionately high in disturbed habitat likely reflect competitive growth and prolific seed production by this disturbance-driven, shade-intolerant species (*Geldenhuys, 2004*) in relatively open, disturbed environments. Seed densities of *V. divaricata* that have been comparatively high in forest-fynbos ecotone and disturbed habitat are in line with this species' ecological role as forest precursor in disturbed or burnt forest margins (*Coetsee & Wigley, 2013*). Low seed densities observed in *A. melanoxylon* across habitat types may partly result from bird dispersal (*Richardson & Kluge, 2008*) and prolonged (since 1986) exposure to a highly effective seed-boring weevil (*Melanterius acacia* Lea) as a biocontrol agent (*Dennill et al., 1999*; *Impson, Hoffmann & Kleinjan, 2009*). In contrast, biocontrol on *A. mearnsii* in the form of a seed-boring weevil (*Melanterius maculatus* Lea) is more recent (introduced in 1993) and only moderately effective, while the more effective flower-galling midge (*Dasineura rubiformis* Kolesik; introduced in 2001) is still localised (*Adair, 2004*; *Impson et al., 2008*; *Impson, Hoffmann & Kleinjan, 2009*). Relative to other invasive acacias in South Africa (*Milton & Hall, 1981*; *Richardson & Kluge, 2008*), the seed bank densities of *A. mearnsii* can be considered moderate to high (similar to *Acacia longifolia* (Andr.) Willd. and *Acacia saligna* (Labill.) H.L. Wendl.), those of *V. divaricata* low to moderate (similar to *Acacia cyclops* A.Cunn. ex G. Don), and those of *A. melanoxylon* the lowest recorded for invasive acacias.

Seed bank densities within species were largely comparable among soil depth classes, with all species displaying a non-significant trend of having fewer seeds in the litter layer than in the shallow and deep layers (most pronounced in *A. melanoxylon*). This trend may in part be due to high levels of seed predation and lateral drift in the litter layer (*Holmes, 1990*; *Richardson & Kluge, 2008*). Seed densities in the shallow and deep soil layers were comparable in our study, although *Richardson & Kluge (2008)* found that seed bank densities in Australian acacias rapidly decline below a depth of 8 cm. The presence

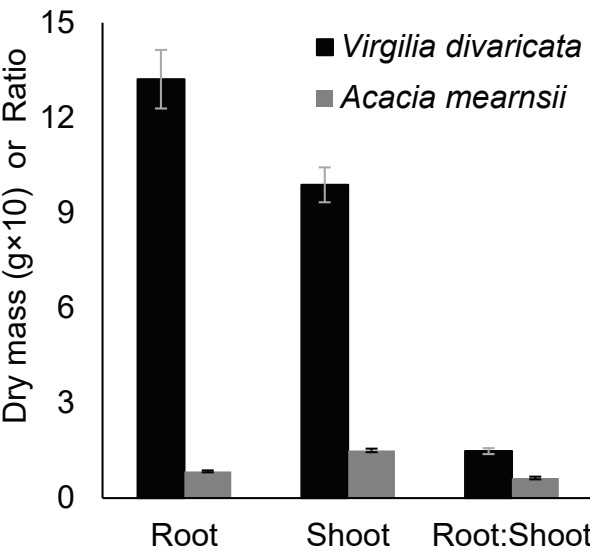

**Figure 3** **Root and shoot dry masses, and root:shoot ratios of *Virgilia divaricata* ($N = 96$) and *Acacia mearnsii* ($N = 110$) seedlings harvested three months after the planting of seeds.** Means ± SE of untransformed values are shown for all measures. The species differed significantly from each other ($p < 0.001$) in terms of all three measures.

of substantial seed densities in deeper soil layers has important management implications (see below).

Seed viability was comparably high (87–91%) in all three study species, and corresponded well with ranges reported in the literature (Table 2) and viability observed for *A. cyclops*, *A. longifolia*, and *A. saligna* (*Milton & Hall, 1981*; *Richardson & Kluge, 2008*). Viability did not differ significantly among layers in the soil profile, as expected of species with long-lived dormant seeds (*Holmes, 1989*). *Acacia melanoxylon* appeared to have the most variability in viability among layers. In characterising the study species' seed ecology, our findings confirmed that *A. mearnsii* exhibited 'Type A' traits (*O'Dowd & Gill, 1986*; *Richardson & Kluge, 2008*), in that large seed banks (Table 2) signified rapid accumulation rates, while equally viable seeds in deep and shallow soil layers (Fig. 2B) signified persistent water-impermeability of seed coats; and that *A. melanoxylon* exhibited 'Type B' traits, in that small seed banks signified gradual accumulation of seeds, while lower seed viability in deep than in shallow soil layers signified more permeable seed coats. We consider *V. divaricata* to have 'Type A' traits given its relatively large seed banks, comparable seed viability in different soil layers (~high dormancy), and occurrence in a fire prone habitat. An opportunistic field survey (Data S1) showed that comparably small proportions of the soil-stored seed banks of *V. divaricata* (19%) and *A. mearnsii* (14%) germinated after a fire (Wilcoxon matched pairs test $Z = 0.511$, $p = 0.609$). Fire-stimulated germination of only a small proportion of the seed bank due to high physical dormancy provides further evidence for 'Type A' seed ecology in both species.
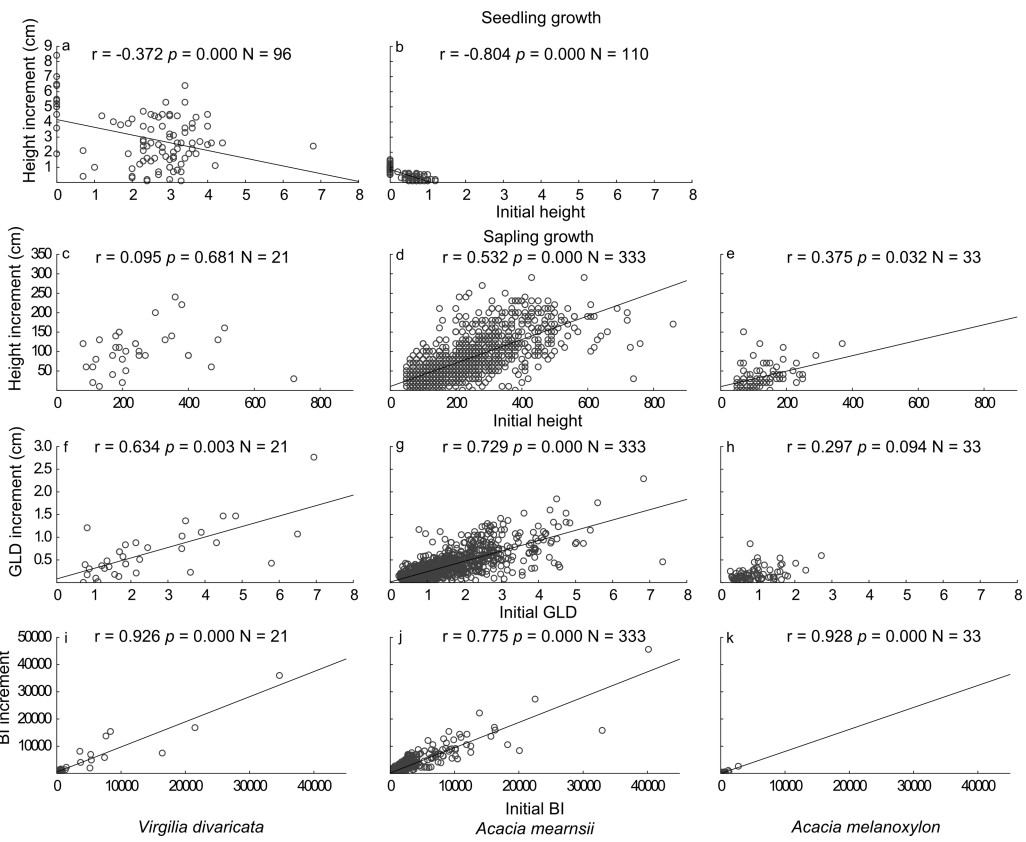

**Figure 4** **Relationships between growth increment and initial size for seedlings and saplings of *V. divaricata*, *A. mearnsii*, and *A. melanoxylon*.** The different panels show seedling growth in height for (A) *V. divaricata* and (B) *A. mearnsii*; sapling growth in (C, D, E) height, (F, G, H) groundline diameter (GLD), and (I, J, K) biomass index (BI) for *V. divaricata*, *A. mearnsii* and *A. melanoxylon*, respectively. Saplings (of 0.2–8.0 cm groundline diameter) regenerated naturally where pine plantations were clear-felled 3–5 years prior to the study.

## Growth performance

The second apparent difference among the study species was that *V. divaricata* significantly outperformed *A. mearnsii* in the seedling stage in terms of root and shoot dry masses (Fig. 3) and growth in height (Figs. 4A & 4B). The superior performance of *V. divaricata* seedlings may relate to its larger seeds providing the seedlings with larger cotyledons (larger initial photosynthetic area) and more rapid hypocotyl extension due to increased seed reserves (*Leishman et al., 2000*). Seedlings of the closely related *V. oroboides*, originally thought to be the same species as *V. divaricata* (*Coates Palgrave, Drummond & Moll, 2002*), similarly outperformed those of two other alien species, *A. longifolia* and *Albizia lophanta* (*McDowell & Moll, 1981*). *Virgilia divaricata* seedlings furthermore invested more resources in root development than in aboveground development relative to *A. mearnsii* seedlings.

Our study of relative growth rates revealed some important differences between *V. divaricata* and *A. mearnsii* suggesting that they utilise divergent growth strategies during the seedling and sapling stages (*Kozlowski, 1971*). However, *A. mearnsii* showed significantly

more positive growth in height relative to initial sapling height than *V. divaricata* (and *A. melanoxylon*), while relative growth rates were similar among species in terms of GLD and BI. Overall, *V. divaricata* with its larger seeds invested in and outperformed *A. mearnsii* in terms of growth in height during the seedling stage, but the opposite occurred during the sapling stage. To establish which of these species would be the best performer overall would require long-term assessments of their growth. However, *A. mearnsii* was found to be more efficient at N resorption and better adapted to low-nutrient environments (characteristic of the study area) than *V. divaricata* (*Van der Colff et al., 2017*).

A third, albeit unexpected finding, was that competitor biomass had no recorded negative effect on the relative growth in BI in focal saplings of any species. Considering that the majority of competitors (and thus competitor biomass) in this study consisted of *A. mearnsii,* this result suggested that *A. mearnsii* does not have an apparent negative effect on growth of *V. divaricata* saplings, even at high densities of the former. This suggests the absence of allelopathic effects of *A. mearnsii* on *V. divaricata* during the sapling stage, congruent with the finding that *A. longifolia* did not inhibit *V. oroboides* germination although the opposite occurred (*McDowell & Moll, 1981*). *Acacia melanoxylon* rarely had numerous or substantially sized saplings occurring in plots, making interpretation involving this species less meaningful.

## Conservation implications

Overall, our findings suggest that amongst the measures considered, the success of *A. mearnsii* as an invader is primarily attributable to its large seed banks, and to a lesser extent its vigorous growth in height as saplings. However, the superior growth performance of *V. divaricata* seedlings and no apparent negative effect of competition from the acacias on sapling growth show promise for its use in integrated management of the acacias. For small, localised infestations, the clearing of acacias can be followed by seeding with pre-treated (chipped or hot water exposed; *Goets, Kraaij & Little, 2017*) seeds of *V. divaricata*, particularly in habitats such as forest margins where *V. divaricata* would occur naturally. Our results suggest that *V. divaricata* saplings are likely to grow well even if interspersed with dense acacias. If *V. divaricata* is afforded a period in which dense stands could establish, this would also promote forest re-establishment, while subsequent germination and growth of the acacias would be suppressed. Such an approach in invaded fynbos shrublands may however in the long-term result in replacement of remaining fynbos by forest species (*Coetsee & Wigley, 2013*). The leaf litter of *V. divaricata* may furthermore play a role in the prevention of *Acacia* germination, as *A. longifolia* germination was found to be reduced under *V. oroboides* litter but not vice versa (*McDowell & Moll, 1981*). The similarities observed between the three species in terms of seed viability and responses to germination stimuli (including timing of germination; *Goets, Kraaij & Little, 2017*) would mean that management operations (such as fire) that would benefit *V. divaricata* recruitment would benefit the acacias equally and thus reduce its efficacy. In addition to the high densities of *A. mearnsii* seeds occurring within the shallow soil layer (i.e., at effective germination depth; *Pieterse & Boucher, 1997*), the presence of equally high densities of seeds in deeper soil may render attempts at depleting the soil seed bank even more challenging.

At a local scale, repeated fires may be used to deplete the shallower soil seed bank (*Auld & Denham, 2006*; *Pieterse & Boucher, 1997*; *Pieterse & Cairns, 1988*), but the rapid progression of *A. mearnsii* seedlings to reproductive maturity (∼two years; (*Praciak et al., 2013*) may result in additional seed production before an adequate fuel load exists. Alternatively, soil solarisation (*Cohen et al., 2013*) may be employed to deplete shallow seed banks, whereas soil inversion (*Holmes & Moll, 1990*) would be required to deplete deeper seed banks. At larger scales, management should rather attempt to prevent the germination of *Acacia* seeds by limiting disturbance events that remove aboveground vegetation, including fire. In fire-prone fynbos, managers should aim for fire return intervals toward the upper end of the ecologically acceptable range (*Kraaij, Cowling & Van Wilgen, 2013b*). Given the smaller seed banks, but bird dispersed seeds of *A. melanoxylon*, new recruitment subsequent to clearing operations needs to be monitored over larger areas than in the case of *A. mearnsii*. To further decrease *Acacia* success, biological control, including seed-boring weevils, flower-galling midges, and a rust fungus (*Impson et al., 2008*; *Impson, Hoffmann & Kleinjan, 2009*; *McTaggart et al., 2015*) should be widely distributed.

## CONCLUSIONS

Seed bank densities were significantly higher in *A. mearnsii* than in *V. divaricata* and *A. melanoxylon* (in decreasing order of abundance), while seed viability was comparatively high in all three species. Seed densities and viability did not differ significantly among soil depth layers, suggesting strong seed dormancy in all three species. *Virgilia divaricata* outgrew *A. mearnsii* in the seedling stage, but during the sapling stage *A. mearnsii* outgrew *V. divaricata* in height. Sapling growth in all species appeared uninfluenced by the extent of competition posed by neighbouring saplings. These findings support the notion that IAPs do not necessarily outperform native counterparts in the invaded environment when exposed to similar conditions (cf. *Daehler, 2003*; *Funk et al., 2008*; *Gioria & Osborne, 2014*) and that *V. divaricata* in some respects (outlined above) shows potential for use in integrated management of the acacias. We recommend that competition among the study species be further investigated in more controlled and longer-term approaches, for instance planting pre-germinated seeds at particular spacing and densities to allow for investigation of the effect of species dominance on competitive interactions across all life stages.

## ACKNOWLEDGEMENTS

We thank South African National Parks and Mountain to Ocean Forestry for permitting sample collection and the establishment of study plots on land under their jurisdiction. We thank Jeanette Pauw for advice with statistical analyses, Benjamin Wigley and Jaco Barendse for their involvement in planning the seedling study, and the following people for assistance with data collection: Marius Strydom, Aletta Botha, Zanri Schoeman, Tiaan Strydom, Gert Botha, Willem Matthee, Kara Marais, Matthew Barnard, Evidence Siwela, Thembisa Sibhakabhaka, Melda Goets, and Christopher Brooke.

## Funding

This work was supported by Nelson Mandela University, Fairfield Tours, the National Research Foundation (South Africa), and Umenzi Colours. The funders had no role in study design, data collection and analysis, decision to publish, or preparation of the manuscript.

## Grant Disclosures

The following grant information was disclosed by the authors:
Nelson Mandela University.
Fairfield Tours.
The National Research Foundation (South Africa).
Umenzi Colours.

## Competing Interests

The authors declare there are no competing interests.

## Author Contributions

- Stefan A. Goets conceived and designed the experiments, performed the experiments, analyzed the data, contributed reagents/materials/analysis tools, prepared figures and/or tables, authored or reviewed drafts of the paper, approved the final draft.
- Tineke Kraaij and Keith M. Little conceived and designed the experiments, analyzed the data, authored or reviewed drafts of the paper, approved the final draft.

## Field Study Permissions

The following information was supplied relating to field study approvals (i.e., approving body and any reference numbers):

Approval was obtained from South African National Parks and MTO Forestry to conduct field studies on land under their jurisdiction.

## Data Availability

The raw data are provided in the Supplemental File.

## Supplemental Information

Supplemental information for this article can be found online at http://dx.doi.org/10.7717/peerj.5466#supplemental-information.

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
