# Peer review of "Seed bank and growth comparisons of native (Virgilia divaricata) and invasive alien (Acacia mearnsii and A. melanoxylon) plants: implications for conservation"

_PeerJ, doi:10.7717/peerj.5466_

## Round 0.1 · original submission · Major Revisions

Dear authors,

Thanks for submitting your research to PeerJ. Your paper is interesting and fall completely within the scope of the journal. I have now received the response of 2 external reviewers. Based in their complete review reports, my decision is that your manuscript is not acceptable for publication in its current form. Below you can find the reports from the reviewers and myself, with some major concerns. Your manuscript must be subjected to an in deep major revision. The reviewed version will be considered again for publication. If you are able to consider ALL the comments below, please review your manuscript accordingly and upload with your reviewed version a detailed “Revision notes” document with an itemized list of specific responses to all the comments and suggestions posed by the reviewers in their letters and annotated pdfs, and by myself. Please, in your reviewed manuscript, highlight the edited and new text in blue font for my convenience. I am convinced that a further effort will improve the quality and clarity of your contribution, there is room for that.

Yours sincerely,
Luis Sampedro
Academic Editor, PeerJ


Academic Editor specific comments
1) Improve the clarity of the experimental design in the methods section, being more specific and precise (in instance including the information of number of sampling sites @L141 and so on).
2) In the results section, please provide the full anova tables with the effects (df of the effect, df of the error term, F and P) of species, soil deep and explicitly those for the interaction. Strongly suggest including the effect of habitat (with the four habitats you sampled) in your statistical model, likely as a random effect.
3) Please edit and improve all the figure captions and table legends, which should be auto-explicative. Please make an effort in improving the information, readability, the quality and clarity of them. In instance, for Fig. 1: For the caption, I would write something than: “Density (a) and viability (b) of seeds found under the canopy of mature trees in natural stands of three species sampled along the coastal plateau of the Cape Coast of South Africa, in the litter, shallow and deep soil layers. Mean ± 95% confidence limits are shown. Sampled trees were 20-50 cm DBH, N = 10.” In the Figure, please show the back-transformed values of means and confident intervals for the panel a. In the axis include the units (seeds m-2 or whatever). In panel b, “Seed viability”. The full anova table of the effects (df of effect and error terms, F and P) of species, soil deep and sps*soil deep must be showed along the figures or alternatively in separate new tables. The same thing applies for the legends of the other figures and tables.
4) In fig 3 and 4 remove the lines of not significant relationships and provide in the figure panels the r, P and N.
5) Rewrite and edit the discussion section for being more precise and specific with your comments, as pointed out by the reviewers.
6) Joint the material in the current section “conclusions” to the general discussion and write a brand new 5-6 lines length conclusion section. Please avoid speculative conclusions and restrict only to those based on your results.

·

Basic reporting

No comment. Please see general comments.

Experimental design

No comment. Please see general comments.

Validity of the findings

No comment. Please see general comments.

Additional comments

Title: Seed bank and growth comparisons of native (Virgilia divaricata) and invasive alien (Acacia mearnsii and A. melanoxylon) plants: implications for conservation.
The manuscript is devoted to study the performance of two invasive alien trees in South Africa, Acacia mearnsii and A. melanoxylon, with that of an indigenous analogue, Virgilia divaricate. Authors analysed soil seed bank (seed densities, their vertical distribution, and viability) and growth rates (of seedlings and saplings), to determine potential for the native species to be used in management of the two invasive plants.
The manuscript include sufficient introduction and background to demonstrate the work fits into the plant invasive field of knowledge. Relevant prior literature are appropriately referenced.
At the end of introduction section authors state the aim of the study. The most important thing is that they seek to make a preliminary study and the data obtained support this objective. Said this, I consider the structure of the manuscript is not in adequate format:
The experimental design is so complex and the description does not allow a good understanding. I can understand the difficulty of explaining this kind of design. I suggest it be accompanied by an explanatory figure.
Figures accompanying the results are not self-explanatory. I miss a complete description in the figure legend as a description of the statistical treatment, meaning of ranked values and the representation of the statistical data in the graph, using letters and asterisks.
I have not clear why authors perform ANOVA in seed bank data and GLMs in seedling and sapling data.
In my honest opinion, I think authors argue largely in the conclusions section (it is not appropriate and not justified) and discussion section is so poor. I suggest merging it and writing a new and appropriate conclusion focused on the stated objective before and limited to the data supported by the results. When you rewrite this section, be careful of too speculative or inaccurate phrases like:
Pg. 13 Line 277-278 ‘Our estimates of seed bank densities were largely consistent with previous investigations (Table 1) and mirrored differences in seedling densities observed for the two acacias (Greet 2016; Tererai et al. 2015).’. Table 1 does not reflect this. The values of the literature are to a large extent higher than those presented in this study. This two references present indirect data about Acacia mearnsii and not for Acacia melanoxylon, by other hand the survey is in different ecosystems and variability, as was showed in this study, to high. Then this sentence is too speculative.
Pg. 15 Line 374-375 ‘The leaf litter of V. divaricata may also play a role in the prevention of Acacia germination, as A. longifolia germination was found to be reduced under V. oroboides litter but not vice versa’. It is too speculative for a conclusion. It would be in the discussion section and should be identified as such.
Pg. 16 Line 376-377 ‘To further decrease Acacia success, biological control (seed-boring weevils, flower-galling midges, and rust fungus; Impson et al. 2008, 2009; McTaggart et al. 2015) should be widely distributed. It is too speculative for a conclusion. It would be in the discussion section and should be identified as such.

Reviewer 2 ·

Basic reporting

I would suggest few improvements in the introduction. Firstly, would be interesting show clearly why the comparison among this species is important. Please, show how this comparison could improve the knowledge about biological invasion, competition, or plant management. Maybe transform your objective from comparison to hypothesis may help to achieve this recommendation.
You addressed competition in the introduction and I agree completely that your approach can contribute effectively in this field. However, I miss an explicit link between competition and the comparison among species performance. Growing and biomass are strongly related with competitiveness of plants and this could be a good way to conduct this link. Additionally, this link will also help to show the need to compare the performance between native and invasive species.
I would like to highlight the need going further in the discussion and explore better some references. You have a good example about how to use the reference to extrapolate your finds (line 374).
Several comments were done in the pdf file and will help the review process.

Experimental design

The methodology is appropriate, but few improvements are needed to make clearer. Details are given in the pdf file.

Validity of the findings

Conclusions are interested and have practical implications. However, few changes could increase the focus in the objectives of the study. Details are given in the pdf file.

Additional comments

Please, check the comment left in the Supplementary material S1.

Annotated reviews are not available for download in order to protect the identity of reviewers who chose to remain anonymous.

---

## Round 0.2 · Minor Revisions

Dear authors,

First of all I have to apologize because the delay handling your manuscript. It has been my fault due some unexpected personal issues.
I am very happy with how you addressed the questions posed in the first round of review. Congratulations. However I would ask you for a further effort, as I agree with the reviewer 2 that there is room for improvements with few effort. Thus my decision is that your paper would be acceptable after minor revision. Please take into account the comments by reviewer 2 (both text and the edited manuscript) for producing a new revised version.

Your sincerely

Reviewer 2 ·

Basic reporting

The authors built an interesting article comparing native and invasive species performance and conducted the discussion in a good way. Especially, the management topic provides information and alternatives to control and eradication of invasive acacias. Nevertheless, few improvements are needed in the article as you can see in the following recommendations and in the PDF file.

The introduction can be improved by adding a sentence in the first paragraph and a new paragraph with focus on management.

Regarding scientific names, look at the lines 176, 249, 310, and 325. I just recommend using the abbreviated names except for the first time in the paper and at the beginning of the sentences.

The supplemental material and the provided raw data is adequate. However, report the geographic coordinates datum in the header and withdraw the comment in the last line in S1.

Experimental design

The sampling design is clear and understandable when we use the Supplemental 2. Therefore, consider using this schematic drawing as a figure. Additionally, I would recommend giving the clear information whether all saplings within the one-meter sub-sample are considered as a focal sapling (between 187 and 189).

Validity of the findings

The section “Conservation implications” is very interesting. Nevertheless, a short change in the text order could reinforce the authors' suggestions about V. divaricate in the lines 349-350.

Additional comments

Could V. divaricate be a problem if used to control biological invasion since “Under natural conditions, V. divaricate typically occurs in dense monospecific stands along forest edges (forest-fynbos ecotone) after fire”? I recommend approaching this topic in the discussion (around lines 366-372, maybe).

Annotated reviews are not available for download in order to protect the identity of reviewers who chose to remain anonymous.

---

## Round 0.3 · accepted · Accept

Dear authors,

Your article has been improved in the two rounds of review and it has been accepted for publication. Thanks for submitting your research to PeerJ. Thanks for your effort. Best wishes,

Luis Sampedro
Academic editor of PeerJ

#